# VIVALDI-CT shaping care home COVID-19 testing policy: A pragmatic cluster randomised controlled trial of asymptomatic testing compared to standard care in care home staff

Oliver Stirrup[1], James Blackstone[2], Iona Cullen-Stephenson[2], Robert Fenner[2], Natalie Adams[3], Ruth Leiser[4], Maria Krutikov[5], Borscha Azmi[5], Nick Freemantle[2], Adam Gordon[6], Martyn Regan[7], Martin Knapp[8], Lara Goscé[1,9], Catherine Henderson[10], Susan Hopkins[3], Arpana Verma[7], Jackie Cassell[3,11], Dorina Cadar[11], Tom Fowler[3], Andrew Copas[1], Paul Flowers[4], Laura Shallcross[5,12]*

1 Institute for Global Health, University College London, London, United Kingdom, 2 Comprehensive Clinical Trials Unit, University College London, London, United Kingdom, 3 UK Health Security Agency, London, United Kingdom, 4 School of Psychological Sciences and Health, University of Strathclyde, Glasgow, United Kingdom, 5 Institute of Health Informatics, UCL, London, United Kingdom, 6 School of Medicine, University of Nottingham, Nottingham, United Kingdom, 7 School of Health Sciences and Manchester Academic Health Sciences Centre, University of Manchester, Manchester, United Kingdom, 8 Department of Health Policy, London School of Economics and Political Science, London, United Kingdom, 9 Department of Infectious Disease Epidemiology, London School of Hygiene & Tropical Medicine, London, United Kingdom, 10 Care Policy and Evaluation Centre, London School of Economics and Political Science, London, United Kingdom, 11 Brighton & Sussex Medical School, University of Brighton, Brighton, United Kingdom, 12 NIHR Biomedical Research Centre at University College London (UCL) Hospital NHS Trust, London, United Kingdom

* l.shallcross@ucl.ac.uk

## Abstract

### Background

Non-pharmaceutical interventions were used widely in care homes for older people during the COVID-19 pandemic, but there have been few randomised trials to support policy decisions. We aimed to evaluate the effect of biweekly asymptomatic staff testing with support funding for sick pay and agency staffing on the clinical outcomes of residents.

### Methods

We conducted a cluster randomised unblinded superiority trial, aiming to recruit up to 280 residential and/or nursing homes in England providing care to adults aged >65 years. Homes were randomised 1:1 to the control arm, which followed national testing policy (comprising symptomatic plus outbreak testing at trial initiation) or intervention (additional twice weekly asymptomatic staff testing for SARS-CoV-2, staff sick pay and agency backfill). Outcomes were evaluated using health data from routine national datasets in combination with aggregate data from participating homes.

**Data availability statement:** Individual-level source data cannot be openly shared, as consent for this was not obtained from study participants. The source data for the study includes potentially identifying information on demographic characteristics and residence within specific care homes, along with sensitive personal medical data. Ethical and legal approval were not provided for onward sharing of individual-level data by the Research Ethics Committee and the Confidentiality Advisory Group. Enquiries regarding data access can be sent to The Comprehensive Clinical Trials Unit at UCL (cctu.vivaldi@ucl.ac.uk), and any applications to make use of the data would be reviewed by a Data Access Committee and require additional ethical approval. However, weekly aggregate data on primary and secondary outcomes at the level of each care home are provided as a Supporting Information file.

**Funding:** This work was supported by the NIHR Health and Social Care Delivery Research (HSDR) Programme (number 154310, L.S. CI, https://fundingawards.nihr.ac.uk/award/NIHR154310). Costs associated with SARS-CoV-2 testing service were funded by the UK Health Security Agency (UKHSA, https://www.gov.uk/government/organisations/uk-health-security-agency). LS is supported by a NIHR Research Professorship (NIHR 302435) and by the NIHR Biomedical Research Centre at UCLH. M.K. was supported by Wellcome Trust Clinical PhD Fellowship award (222907/Z/21/Z, https://wellcome.org/). The main study funder (NIHR) had no role in study design, in the collection, analysis and interpretation of data, in writing of the report, or in the decision to submit the paper for publication. UKHSA staff members contributed to the study design, interpretation of data and writing of the report, but the decision to submit for publication was predetermined as part of the project plan. Decisions regarding the continuation and reporting of the trial were made by the independent Trial Steering Committee.

**Competing interests:** I have read the journal's policy and the authors of this manuscript have the following competing interests: L.S. reports grants from the Department of Health and Social Care during the conduct of the study and is a member of the Social Care Working Group, which reports to the Scientific Advisory Group for Emergencies. All other authors report no conflicts of interest.

The primary outcome was the incidence of COVID-19-related hospital admissions in residents.

## Results

The trial was conducted from January to August 2023, with 41 care homes randomised to intervention and 40 randomised to control included in the analysis. The trial was stopped early as it was not adequately powered for the primary outcome due to site recruitment and primary outcome events being substantially lower than expected. There was no significant difference in the primary outcome of resident COVID-linked hospital admission incidence between intervention and control arms (22.7/1000 person-years vs 15.0/1000 person-years, incidence rate ratio 1.19, 95%CI 0.55–2.58, P = 0.66; incidence rate difference 4.0, 95%CI −14.3 to 22.2). Trial set up took less than three months. Most trial outcomes were derived from routinely collected data. Recorded uptake of staff testing in the intervention arm was low (mean per home each week 14.4%).

## Conclusion

This trial was not well-powered to evaluate the impact of the intervention on the primary outcome, and recorded uptake of staff testing was low. However, our pre-existing care home network underpinned by linked routinely collected data provides a model for more agile interventional studies in the care home setting.

## ClinicalTrials.gov registration

NCT05639205.

---

## Introduction

The COVID-19 pandemic led to a stark increase in care home resident mortality in many countries, which arguably justified the rapid deployment of stringent public health disease control measures [1]. However, in the aftermath of the pandemic, concerns have been raised about the hidden human cost for residents, families and staff of some of the measures that were used to prevent the spread of infection [2,3]. These measures impacted residents' independence and dignity and restricted their ability to participate in social activities [4,5].

Care home residents represent a substantial proportion of society, with nearly 280,000 people aged > 65 years recorded as living in a care home in England and Wales in the 2021 national Census [6]. The majority of care home residents are aged >85 years [6], at least two-thirds live with dementia, and over half die within 12 months of admission to a care home [7]. Non-pharmaceutical interventions (NPIs) such as cohorting residents, visitor restrictions, symptomatic testing and face masks are widely used in care homes to curtail the spread of outbreaks of respiratory and gastrointestinal infections [8,9]. During the COVID-19 pandemic, these standard NPIs

were supplemented by novel additions such as UK central government-funded regular asymptomatic testing and sickness pay for care home staff [10–13]. The evidence base to support the use of these additional NPIs in care homes is limited, requiring extrapolation of findings from healthcare settings and with few examples of interventional studies evaluating the use of NPIs against COVID-19 or other infections [14–16]. Some evidence from observational studies suggests a mitigation of the impact of COVID-19 on mortality among care home residents resulting from higher levels of asymptomatic testing for SARS-CoV-2, particularly prior to the introduction of COVID-19 vaccines [16,17].

Our research team previously established the VIVALDI observational study of care home residents in England [18]. This study used data-linkage between SARS-CoV-2 testing records, vaccination data and routine hospital admission and mortality datasets to investigate issues such as infection incidence [19], immunology [20] and vaccine effectiveness [12,21,22]. However, the observational design of this study meant it was not well suited to evaluating the impact of testing, which is likely to be confounded by a range of care home characteristics that are difficult to measure, such as uptake of other NPIs.

In 2022, to inform national policy around NPIs in care homes for winter 2023/2024, we planned a rapid, pragmatic, cluster-randomised controlled trial [VIVALDI-CT, ISRCTN13296529] [23]. At this point in time COVID-19 was still circulating and care home staff were still potentially willing to undergo asymptomatic testing if offered. The trial compared the intervention 'Test to Care' comprised of biweekly asymptomatic staff testing with staff sick pay and agency backfill against standard care (i.e., English national testing policy in care homes). The trial ultimately did not recruit its target sample size of care homes due to shifts in the epidemiological and policy context in the UK. In this article we report on the trial outcomes obtained relating to our original aims, and also explore how this study could help inform the development of future cluster trials in residential care homes.

## Methods

VIVALDI-CT was designed as an unblinded cluster randomised controlled trial of care homes for older adults in England, with a 1:1 allocation of individual homes to the intervention or control arms (ClinicalTrials.gov: NCT05639205) [23]. Care homes in the control arm were subject to a national SARS-CoV-2 testing policy; this comprised symptomatic testing plus outbreak testing with no Government funding for sick pay at the start of the trial. The intervention comprised additional twice weekly (voluntary) asymptomatic staff testing for SARS-CoV-2 with lateral flow devices (LFDs), alongside a coproduced communications strategy to encourage testing and support payments for paid sick leave and agency-back fill for COVID-related sick leave. Groups of care homes were recruited through contact with the senior management teams of larger providers in England, although a small number of independent homes were also recruited. All care home staff were eligible to participate in the testing intervention, including temporary staff with no restrictions (e.g., including catering, administrative and maintenance staff), but not professionals visiting the care home, such as GPs and health visitors. All residents of participating care homes were eligible for evaluation of trial outcomes.

The primary outcome was incidence of COVID-19 related hospital admissions in residents. Secondary outcomes relating to residents included incidence rates of all-cause hospital admissions, COVID-associated mortality, all-cause mortality and SARS-CoV-2 infections. Secondary outcomes relating to staff and evaluated at each home on a weekly basis included the proportion of staff participating in testing, the prevalence of SARS-CoV-2 among staff who tested (proportion with positive result among those with at least one test recorded during each week), proportion off sick and proportion of all shifts filled by agency staff. It was planned that secondary outcomes would also include home-level data on incidence rates of SARS-CoV-2 outbreaks and care home closures due to outbreaks, and duration of outbreaks. A secondary composite incidence outcome of COVID-19-linked hospital admission or mortality in residents was added at time of writing the statistical analysis plan, as this would provide greater power for the detection of a difference between trial arms than either outcome alone. Incidence outcomes were measured from 2 weeks after trial initiation at each site up to the point at which either they left the trial of the trial ended.

## Ethics and consent

The study received ethical approval from the London – Bromley Research Ethics Committee (22/LO/0846) and Health Research Authority Confidentiality Advisory Group approval (22/CAG/0165) for data use, both on 19th December 2022. Consent was not required for the analysis of pseudonymised data for staff or residents, but individuals could request to opt-out and have their data omitted from transfer to UCL and subsequent trial analyses. Staff testing was not mandatory.

## Data sources

In order to minimise the burden on care home staff, much of the data for analysis were obtained from routinely collected healthcare information held within the UK COVID-19 Datastore. This included results of LFD and Polymerase Chain Reaction (PCR) tests for SARS-CoV-2 and information on resident hospital admissions (UK National Health Service's Hospital Episode Statistics) and deaths (UK Office for National Statistics). Data within the COVID-19 Datastore are linked to a pseudonymised ID code for each individual, which could be linked to unique codes (Care Quality Commission IDs (CQC-IDs)) for participating care homes and associated staff or resident status through SARS-CoV-2 test records. We carried forward data-linkage of residents to specific homes for 1 year from the date of their latest SARS-CoV-2 test (whether or not this predated the trial period). Demographic data including age, sex, care home, and care home role were available from SARS-CoV-2 testing records which were entered by requestor during the test registration process (self or by care home staff member).

Data on hospital admissions and deaths are linked to ICD10 diagnostic codes within the COVID-19 Datastore. However, there is a lag of several months in the assignment of ICD10 codes to hospital admission data. To allow timely monitoring of the primary outcome and avoid the risk of missing relevant admissions, care providers were asked to upload records of COVID-associated hospital admissions at participating care homes to the COVID-19 Datastore. These data were also linked to pseudonymised ID codes for each individual.

The management teams of care home providers were asked to provide weekly aggregate data for each participating care home including the total number of residents, total number of staff and number of staff opting out of asymptomatic testing (for intervention arm), staff sickness absence and employment of agency staff, and number of residents with COVID-19-linked hospital admission (i.e., the sum of primary outcome events). Data from the UK COVID-19 Datastore were processed in the secure NHS Foundry system, within which relevant data records were identified for participating care homes and opt-out requests applied. Datasets for analysis were exported from Foundry to the UCL Data Safe Haven (a walled garden secure environment to ISO27001 standards) and collated with the aggregate data from Providers.

## Outcome definitions

The trial team checked provider-supplied data on hospital admissions against routinely collected data prior to final analyses and without viewing intervention allocations. It was confirmed that primary outcome events would be defined, using routine data, as hospital admissions with ICD10 code for confirmed COVID-19 ('U071', not limited to primary code) and/or any admissions in residents who test positive for COVID-19 within 24h following admission or 7 days before. Repeat hospital admissions within 30 days were not included. COVID-associated mortality events were defined as death with any ICD10 code (not limited to primary) of confirmed COVID-19 and/or within 28 days of a positive SARS-CoV-2 test. All analyses of resident outcomes used weekly numbers of care home residents from Providers to inform the 'exposure' variable of person-days at risk, and incidence rates were expressed per 1000 person-years (/1kPY).

## Sample size

Based on observational data from the VIVALDI study [18], we assumed that we would observe a cumulative incidence for the primary outcome of approximately 3.0% in the control arm, requiring a trial duration of 5–6 months, in combination

with a conservative intraclass correlation coefficient (ICC) value of 0.01 and average care home size of 35 residents. We planned to recruit 280 homes randomised 1:1 to trial arms and, using a two-sided test at a 5% significance level, this would provide 84% power to detect a reduction in primary outcome due to the intervention to 1.9% (relative risk 0.63).

### Randomisation and masking

The randomisation process was conducted separately for each Provider joining the study or for batches of homes joining the study from a Provider if required. Restricted randomisation was used to ensure a balance between arms by region and in the mean number of residents per home for each Provider; in this process a set of intervention allocations is selected at random from amongst the subset of possible allocation in which balance is within a pre-specified tolerance level (full details in randomisation plan). Randomisation was conducted by the Trial Statistician (O.S.) using the *cvcrand* package for R.

Staff and residents of participating care homes were not blinded to their intervention allocation. The Trial Statistician remained blinded to intervention allocation up until the final stages of analyses for the interim report. The same analysis code formed the basis for reporting of results after completion of the study. The final decision regarding the coding of the primary outcome based on the different data sources for the final analysis was taken based on a dataset with information on care homes and the associated intervention allocation for each resident removed.

### Statistical analysis

Analyses for all outcomes were carried out on an 'intention-to-treat' basis according to the allocated trial arm of each home. If a Provider or individual care home entirely dropped out of the trial and provided no further data, then we considered the relevant homes to be no longer under follow-up. Analyses were conducted using Stata V18.0.

Incidence outcomes relating to hospitalisations and positive SARS-CoV-2 tests of individual residents (including primary outcome) were analysed on a weekly aggregate basis. Mortality incidence outcomes were analysed on a monthly basis, as only month of death was exported for analysis. Mixed effects negative binomial models, or Poisson models if convergence failed, were used for all these outcomes, with cluster-robust standard errors. Data for incidence outcomes in the first 2 weeks of involvement for each site were considered to represent a transition period for intervention sites and were omitted from the analysis. The main effect estimate was given by incidence rate ratios (IRRs), and the marginal incidence rate difference was also calculated.

For outcomes expressed as a proportion (e.g., staff testing per week), the analysis used mixed effects logistic regression applied to weekly data from each home, with cluster-robust standard errors. The main effect estimate was given by adjusted odds ratios (aORs), with marginal risk difference also calculated.

Analysis models for primary and secondary outcomes included adjustment for care home provider (grouping any independent homes), region, size (number of residents) at study entry, and calendar time. Care home size and calendar time were included as continuous variables for most outcomes, allowing for potentially non-linear adjustment using a 5-knot restricted cubic spline [24]. For mortality outcomes, adjustment for calendar time used a categorical variable for calendar month. For adjustments related to both calendar time and care home size, a simpler relationship (e.g., linear) was considered if the spline model did not converge. All mixed effects models included normal random intercept terms for each care home.

A single interim report prepared by the Trial Statistician was planned after 3–6 months of operating the intervention to allow the trial's Data Monitoring and Ethics and Trial Steering Committees (DMEC and TSC) to assess the emerging benefit: risk ratio, the level of testing in each trial arm and projected power based on site recruitment and primary outcome event rate. The TSC reviewed site recruitment and staff testing data by arm alongside primary and key secondary outcomes in the control arm only, with outcome data by arm also provided to the DMEC. Further details of the planned analyses are given in supplementary S1–S3 Files.

## Results

Eighty-five care homes were recruited, with 43 randomised to intervention and 42 to control (study flow chart shown in Fig 1, as per CONSORT checklist S4 File). Two homes in the intervention arm and two in the control arm dropped out of the study before any outcome data collection. Follow-up for the first sites began on 9th January 2023 (Fig 2). An interim report was presented to the DMEC and TSC in June 2023, and on 5th July 2023, the TSC formally recommended that the trial be stopped as it was not adequately powered to evaluate the intervention's impact on the primary outcome. This judgement was based on lower than planned recruitment of homes and substantially lower than expected incidence of the primary outcome in the control arm. Follow-up ended for the last set of sites on 6th August 2023.

Despite early cessation of the trial, we have reported full trial results as per the study's Protocol and Statistical Analysis Plan. The trial was prospectively registered (ClinicalTrials.gov registration: NCT05639205) and we feel that full reporting of the study's results is appropriate for transparent completion of the research process and to respect the contribution of research subjects (in line with the Declaration of Helsinki Article 36 [25]). Furthermore, full reporting of trial outcomes demonstrates the feasibility of our novel use of routine health data for a randomised trial in the context of residential care homes, which allowed us to minimise the additional workload required from care staff of participating homes; we believe that development of this approach will be crucial to the delivery of greater numbers of randomised controlled trials in this setting, as well as enabling the responsiveness and compressed timelines required for research related to epidemic and pandemic diseases.

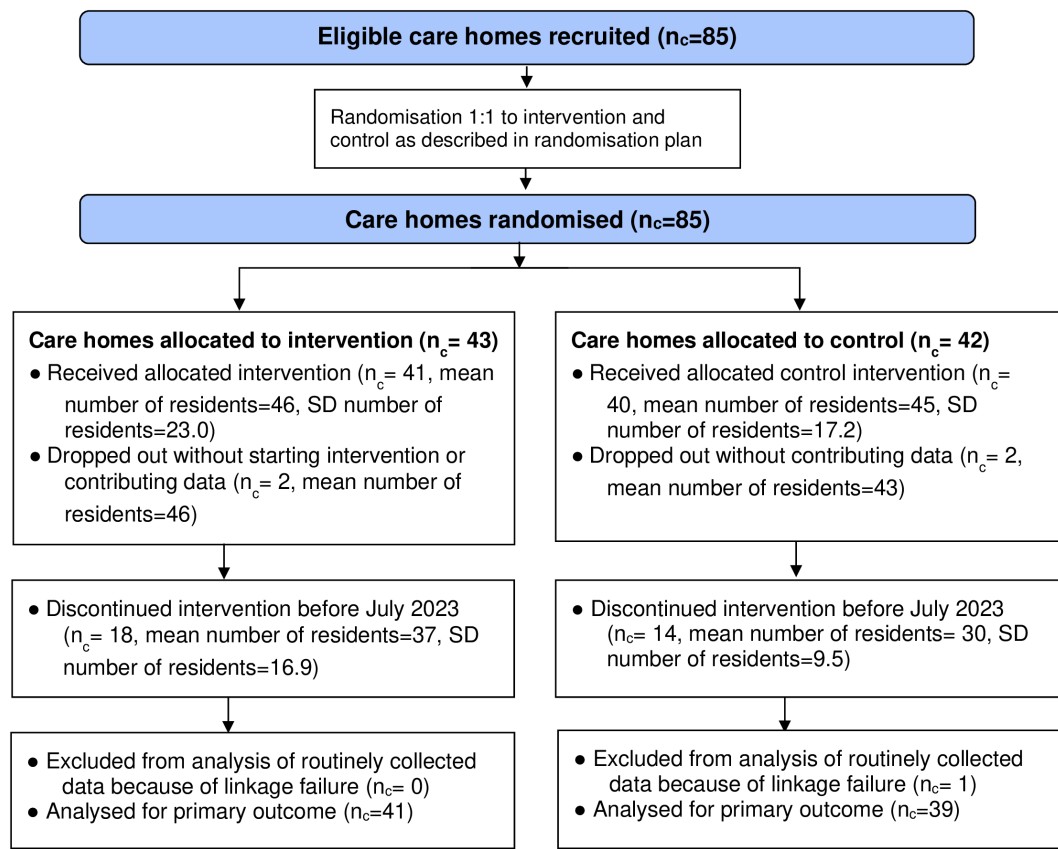

**Fig 1. Flow diagram of study site enrolment and intervention implementation.**

Baseline characteristics of the randomised homes by trial arm are presented in Table 1. Baseline survey data are missing from two of the care homes that were randomised but dropped out before starting data collection. The median number of residents per home was 40 (IQR, 33–55) in the control and 39 (IQR, 29–57) in the intervention arm.

Aggregate data were collected from Providers for 1356/1396 (97.1%) site-study-weeks, with information on resident numbers for missing weeks filled in by interpolation. There was one control arm site for which we were unable to establish linkage to routinely collected data based on the home's CQC-ID; this site was excluded from analysis of all affected outcomes. Linear rather than spline [24] adjustment was used for calendar time for all analysis models, and further model simplifications were required in some cases (S1 Table). Outcomes related to outbreak events were dropped from the analysis as this information was not collected reliably across all regions, and planned sensitivity analyses were also dropped due to the smaller than planned sample size (details in supplementary S1 Appendix). Weekly aggregate data on primary and secondary outcomes at the level of each care home are provided in supplementary S1–S2 Datas.

The mean proportion of staff per home with a recorded SARS-CoV-2 test result each week was 1.3% (median: 0%, IQR: 0–1%) in the control arm and 14.4% (median: 13%, IQR: 3–21%) in the intervention arm. There was a reduction in logged staff testing beyond the end of March 2023 in the intervention arm and near-total cessation in the control arm (Fig 3). This corresponds to a change in the national testing guidelines at the start of April 2023, with an end to routine symptomatic and outbreak testing of staff for SARS-CoV-2. The mean proportion of staff reported by intervention sites to be opting out of testing per home per week was 16.3% (median: 5%, IQR 0–20%).

There was no significant difference in the primary outcome of resident COVID-linked hospital admission incidence between intervention and control arms (22.7/1kPY vs 15.0/1kPY, IRR 1.19, 95%CI 0.55–2.58, P = 0.66, Table 2). The control arm incidence corresponds to an individual-level risk over a 6-month period of 0.75%. A total of 22 primary outcome events were observed using routine health data, but only one of these was directly uploaded to the COVID-19 Datastore as an individual trial event, and Providers reported a total of eight COVID-linked admissions in their weekly aggregate

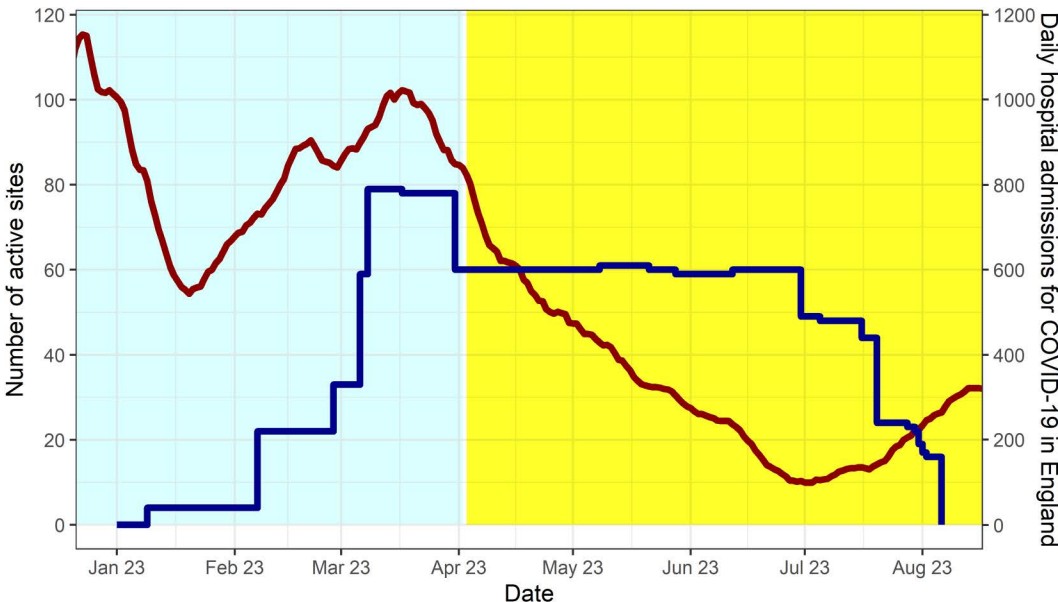

**Fig 2. Stepped graph showing the number of active care homes taking part in the trial over time (dark blue line, left y-axis), along with the daily number of COVID-19 linked hospital admissions in England (red line, right y-axis).** Background shading indicates the change in national staff testing policy from symptomatic plus outbreak testing (light blue) to limited symptomatic testing only for those eligible for antiviral therapeutics (yellow). National admissions data sourced from the UK Government COVID-19 Dashboard, processed by Our World in Data.

**Table 1. Baseline characteristics of the randomised care homes by trial arm.**

| | Trial arm | |
|---|---|---|
| | **Control** | **Intervention** |
| *n* care homes | 42 | 43 |
| Region | | |
| North | 11 (26.2) | 9 (20.9) |
| Midlands | 15 (35.7) | 17 (39.5) |
| South | 14 (33.3) | 12 (27.9) |
| London | 2 (4.8) | 4 (9.3) |
| Care home size (*n* residents) | 40 (33-55) | 39 (29-57) |
| Care home type | | |
| For profit | 35 (83.3) | 36 (83.7) |
| Not for profit | 6 (14.3) | 6 (14.0) |
| Number of permanent staff | 61 (40-76) | 57 (38-77) |
| Type of care, by bed | | |
| Nursing (%) | 26.6 | 24.7 |
| Residential (%) | 32.8 | 45.7 |
| Dementia (%) | 40.7 | 29.6 |
| Bed occupancy (%) | 86.7 | 92.0 |
| Ethnicity of staff | | |
| Asian or Asian British (%) | 9.6 | 13.8 |
| Black, Black British, Caribbean or African (%) | 5.7 | 6.5 |
| Mixed or multiple ethnic groups (%) | 0.9 | 0.6 |
| White British (%) | 42.1 | 44.6 |
| White other (%) | 11.7 | 7.7 |
| Other ethnic group (%) | 0.7 | 1.2 |
| Not reported (%) | 29.2 | 25.5 |
| *n* outbreaks in 3 months prior to study entry | 1 (0-2) | 1 (0-1) |
| Home in outbreak at study entry | 5 (11.9) | 5 (11.6) |
| Disease control measures in place at study start | | |
| Staff wearing masks at work | 6 (14.3) | 7 (16.3) |
| Social distancing protocols for visitors | 3 (7.1) | 4 (9.3) |
| Social distancing protocols for staff | 0 (0.0) | 1 (2.3) |
| Cohorting of staff and residents | 6 (14.3) | 4 (9.3) |
| Enhanced cleaning procedures | 11 (26.2) | 11 (25.6) |
| Proportion of staff with ≥2 vaccine doses (%) | 89 (83-96) | 93 (80-96) |
| CQC rating | | |
| Outstanding | 6 (14.3) | 3 (7.0) |
| Good | 29 (69.0) | 35 (81.4) |
| Requires improvement | 5 (11.9) | 4 (9.3) |
| Inadequate | 1 (2.4) | 0 (0.0) |

Data presented as *n, n* (%), % [aggregated over all care homes] or median (IQR).

data (further details in Appendix); this implies that the use of routine healthcare data has the potential to provide more reliable ascertainment of participant outcomes than trial-specific data collection in the setting of cluster trials in residential care homes.

There was no significant difference between intervention and control arms in the incidence rates among residents of COVID-linked mortality (IRR 0.64, 95% CI 0.15–2.70, P = 0.54), all-cause mortality (IRR 1.13, 0.84–1.52, P = 0.43), the composite event of COVID-linked hospital admission or mortality (0.95, 0.43–2.08, P = 0.89) or in the incidence of SARS-CoV-2 infections (2.65, 0.77–9.19, 0.12) (Table 2). However, there was a statistically significant reduction in the rate of all-cause hospital admission in the intervention arm relative to control (IRR 0.74, 95%CI 0.56–0.98, P = 0.03) (this was also observed on unadjusted analyses, S2 Table).

As expected, the prevalence of SARS-CoV-2 among staff who tested in the intervention arm was lower than that in the control arm (8.1% vs 26.9%, aOR 0.25, P < 0.01). There was no difference between intervention and control arms in the proportion of staff on sick leave each week (6.4% vs 6.2%, aOR 0.99, P = 0.90) or in the proportion of shifts filled by agency staff (6.4% vs 5.1%), aOR 1.25, P = 0.73).

## Discussion

We aimed to evaluate the policy of asymptomatic testing and sickness pay for care home staff to prevent severe outcomes in residents following COVID-19 infection in a pragmatic, cluster randomised controlled trial. By necessity, the trial took place in a rapidly evolving policy and epidemiological context and consequently stopped early when it became apparent that an adequately powered evaluation of the primary outcome could not be achieved. No differences were detected between control and intervention arms for either the primary outcome of COVID-19 related hospital admissions or the pre-specified secondary outcomes of all-cause or COVID-19 related mortality. Although the trial was not ultimately powered to meet its primary objective, we nonetheless feel that the results represent an important contribution to the development of large-scale and high-quality research in the setting of care homes for older people. We also hope that our results might contribute to future systematic reviews and meta-analyses, for which all evidence stemming from randomised studies can be useful [26,27].

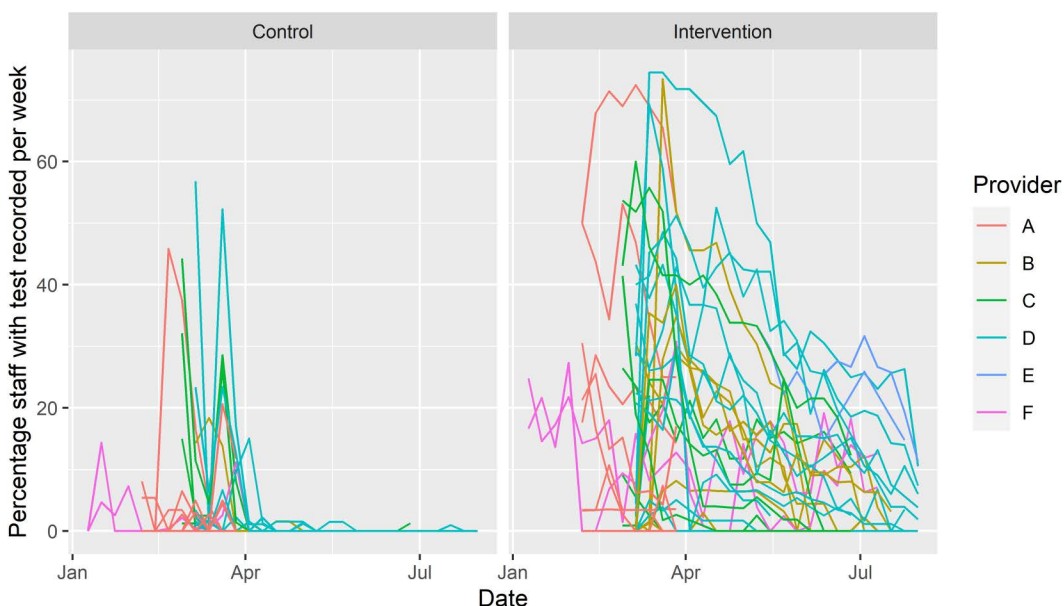

**Fig 3. Line graph of the percentage of staff at each participating home with at least one SARS-CoV-2 test recorded in the national testing dataset within each week of the study.**

**Table 2. Summary of primary and secondary outcomes in control and intervention arms.**

| | Trial arm | | | |
| | Control | Intervention | Intervention vs control | |
| | Median (IQR, range) | Median (IQR, range) | | |
|---|---|---|---|---|
| Weeks of participation per care home | 19 (18-22, 8-27) | 18 (8-19, 2-27) | — | — |
| *Primary outcome* | **Overall [event count]; Median (IQR, range) per home** | **Overall [event count]; Median (IQR, range) per home** | **IRR\* (95% CI, *P*)** | **IRD\* (95% CI)** |
| IR of COVID-19 hospital admissions per 1kPY†# | 15.0 [9]; 0 (0-0, 0-443) | 22.7 [13]; 0 (0-0, 0-513) | 1.19 (0.55-2.58, 0.66) | 4.0 (−14.3 to 22.2) |
| *Secondary outcomes* | **Overall [event count]; Median (IQR, range) per home** | **Overall [event count]; Median (IQR, range) per home** | **IRR\* (95% CI, *P*)** | **IRD\* (95% CI)** |
| IR of all-cause hospital admissions per 1kPY† | 506.6 [293]; 458 (239-754, 74-1240) | 367.3 [204]; 325 (140-548, 0-993) | 0.74 (0.56-0.98, 0.03) | −126.9 (−244.9 to −9.0) |
| IR of COVID-19 mortality in residents per 1kPY† | 8.4 [5]; 0 (0-0, 0-289) | 5.4 [3]; 0 (0-0, 0-65) | 0.64 (0.15-2.70, 0.54) | −3.1 (−13.6 to 7.5) |
| IR of all-cause mortality in residents per 1kPY† | 225.9 [134]; 203 (100-322, 0-646) | 243.4 [136]; 208 (94-342, 0-873) | 1.13 (0.84-1.52, 0.43) | 27.5 (−39.6 to 94.5) |
| Composite IR of COVID-19 hospital admissions and mortality per 1kPY† | 21.9 [13]; 0 (0-46, 0-464) | 26.8 [15]; 0 (0-0, 0-539) | 0.95 (0.43-2.08, 0.89) | −1.5 (−23.6 to 20.5) |
| IR of SARS-CoV-2 infections in residents per 1kPY† | 45.1 [27]; 0 (0-38, 0-3046) | 72.2 [41]; 0 (0-34, 0-1054) | 2.65 (0.77-9.18, 0.12) | 117.1 (−45.9 to 280.2) |
| | **Overall mean; Median (IQR, range)‡** | **Overall mean; Median (IQR, range)‡** | **aOR\* (95% CI, *P*)** | **Marginal % dif.\* (95% CI)** |
| Proportion of staff testing each week (%) | 1.3; 0 (0-1, 0-13) | 14.4; 13 (3-21, 0-65) | 29.15 (10.83-78.49, <0.01) | 16.7 (6.4 to 27.0) |
| Prevalence of SARS-CoV-2 among staff who test each week (%) | 26.9; 6 (0-67, 0-100) | 8.1; 3 (1-7, 0-100) | 0.25 (0.11-0.55, <0.01) | −7.3 (−12.0 to −2.7) |
| Proportion of staff per home off sick each week (%) | 6.2; 6 (4-8, 0-11) | 6.4; 6 (4-9, 0-16) | 0.99 (0.78-1.24, 0.90) | −0.1 (−1.5 to 1.3) |
| Proportion of all shifts filled by agency staff each week (%) | 5.1; 4 (0-7, 0-42) | 6.4; 2 (0-7, 0-68) | 1.25 (0.36-4.35, 0.73) | 1.2 (−5.3 to 7.7) |
| Proportion of staff explicitly opting out of testing each week (%) | — | 16.3; 5 (0-20, 0-99) | — | — |
| Proportion of staff off sick with COVID-19 each week (%) | 0.5; 0 (0-1, 0-3) | 1.0; 1 (0-1, 0-5) | — | — |

aOR, adjusted odds ratio; dif., difference; IQR, interquartile range; IR, incidence rate; IRD, incidence rate difference; 1kPY, 1000 person-years. \*Adjusted for care home provider, region and size and calendar time unless specified otherwise. †Descriptive data and statistical model summaries do not include data from first 2 weeks of trial participation at each site. ‡Median, IQR and range given for aggregate values for each site. #Intraclass correlation coefficient of 0.002 based on linear mixed model.

## Interpretation of trial outcomes

There was a nominally statistically significant reduction in all-cause hospital admissions between control and intervention homes. However, this should be interpreted with appropriate caution because weekly tests were logged by only 14% of staff in intervention homes, implying a level of testing too low to cause an appreciable impact, and the neutral primary analysis means that this result should be considered exploratory. Low uptake of testing was likely due to a combination of testing fatigue and the fact that the salience of the behavioural intervention, which primarily relied on financial incentives

(sickness pay) to encourage staff to test, was reduced by lower than anticipated rates of COVID-19 in the community. There have also been increasing concerns about the adverse impacts of the continuing use of NPIs, more generally, on the physical and mental health of care residents and staff [28].

Despite the vulnerability of the care home population, with the exception of interventions to improve hand hygiene [29–31], there have been few trials of NPIs to reduce transmission of COVID-19 or other respiratory infections in this setting [14,32–35]. Substantial uncertainty remains regarding the effectiveness, cost-effectiveness, benefits, and harms of different disease control measures, and this inevitably contributed to delays in responding to the COVID-19 pandemic. To date, the most comprehensive evaluation of the effectiveness of SARS-CoV-2 testing for staff to prevent infections and severe outcomes in residents within the scientific literature is an observational study using data from 13,424 nursing homes in the USA [16]. In this study, asymptomatic testing was associated with a significant reduction in infections and COVID-19 related deaths in residents in the period before vaccines were available. It had no effect on resident outcomes in the period between vaccination rollout and the emergence of the Omicron (B.1.1.529) variant but was associated with a reduction in rates of infection in residents during the period of Omicron dominance [16]. Whilst this suggests a potential role for asymptomatic testing in staff to protect residents when there is a substantial risk of severe infection-related outcomes, the authors were unable to account for concurrent usage of other NPIs (except personal protective equipment), which makes it difficult to discern the direct effect of asymptomatic testing. This cohort study was also unable to evaluate the potential negative impacts of asymptomatic testing on resident's well-being, physical and mental health. A report commissioned by UKHSA evaluated the SARS-CoV-2 testing pro-gramme in England between October 2020 and March 2022 within adult social care settings [17] and also concluded that higher levels of staff testing were associated with reduced incidence of infections and subsequent deaths among residents. Cost-effectiveness in preventing deaths was found to be highest in the period prior to completion of primary vaccination.

There was a discrepancy between the number of staff at intervention sites who reported opting out of testing based on information provided by the care home (mean 16.3%) and the number of test results that were logged in routinely collected data (mean weekly value 14.4% of staff), with these combined values falling far short of 100%. We do not know whether this was primarily due to staff testing but not logging their results or due to a lack of testing without staff members formally stating that they had opted out. It is likely that behaviours changed over the course of the trial. To the end of March 2022 fewer than 25% of LFDs distributed to care home settings resulted in a reported test result, so under-reporting of LFD use seems to have been a long-standing issue [17]. Care homes were willing to provide weekly aggre-gate data on staff and residents but they appear to have missed a substantial number of COVID-19-related resident hospital admissions when collating these data, and the planned data pipeline for secure upload of individual-level data on these events was not used by most providers.

## Epidemiological context

When our trial was conceived, asymptomatic testing for staff was still recommended nationally in the UK. By the time we started, only those who were symptomatic were eligible for testing, and policy was further revised on 3rd April 2023 to restrict symptomatic testing to only the minority of staff eligible for COVID-19 therapeutics. This change coincided with a marked decline in asymptomatic testing in intervention homes, highlighting the challenges of conducting research in a changing epidemiological and policy context. Hypothetically, it may have been preferable to have undertaken this trial 6–12 months earlier when the incidence of COVID-19 within care homes was higher [36], asymptomatic testing was still recommended as national policy, and the request to care providers would have been to opt out of rather than re-instate asymptomatic testing. This approach may have been more palatable to care home staff, but it would have raised ethical and practical challenges by contravening the public health guidance that was in place at the time of the trial and poten-tially leaving care homes exposed to claims of negligence, for example in the event of an outbreak.

## Lessons learnt

Our successful ascertainment of trial outcomes demonstrates the feasibility of our novel use of routine health data for a cluster randomised trial in the context of residential and nursing care homes. This is an area where more high-quality randomised research is urgently needed to inform policy, but where such studies are very difficult to set up due to difficulties relating to recruitment of care homes [37] and individual residents, and a lack of embedded research staff and infrastructure. A key barrier to the delivery of any kind of research in care homes is a lack of staff capacity to support study delivery because staff are overstretched with high vacancy rates, and research is not typically part of the role for front-line carers. In England, studies conducted in the NHS benefit from support and direct financial incentives for recruitment via the NIHR Research Delivery Network, but this infrastructure is undeveloped in care homes. Cluster trials typically require large number of individual participants [38], making it very challenging to collect all of the data needed for a trial. The creation of a reliable pipeline for routine healthcare data of care home residents is therefore crucial for the development of future large-scale trials in this setting in the UK.

The preceding Vivaldi observational study (ISRCTN 14447421) began as a SARS-CoV-2 sero-prevalence study [18], but we realised that there was a unique opportunity to support the COVID-19 response in care homes by integrating routinely collected individual-level testing and vaccination data from residents and staff, linked to healthcare outcomes. This made it possible—for the first time ever in England—to establish an accurate database of care home residents and staff that could be used for COVID-19 research to directly inform policy in near real-time. The resulting cohort study included >300 care homes and individual-level data from >70,000 staff and residents, which we used to generate evidence on disease burden and severity [36], variants [39] and vaccine effectiveness [12,21]. For the present trial, we re-purposed the care home network and data infrastructure established through Vivaldi to rapidly set-up a pragmatic, cluster randomised trial.

We are developing a new pilot study, Vivaldi Social Care [40], with the goal to make it easier for care homes to participate in research studies (by keeping the workload associated with study participation to a minimum), to increase research impact and efficiency by enabling studies to progress at pace and scale, and to deliver research in partnership with the care sector. We are expanding the Vivaldi network to c.1500 care homes, and establishing new processes to access infection-related routinely collected data from residents in these homes. Through collaboration with care sector organisations (e.g., The Outstanding Society, Care England and others), and extensive engagement with residents, relatives, charities and care homes staff, we are gradually changing the culture and understanding of research in homes that are participating in this project. A key technical development for the Vivaldi Social Care project [40] is that the data pipeline will be based on identification codes (i.e., NHS numbers) of residents exported daily directly from the electronic record systems of participating care homes; this will provide an improved level of accuracy regarding admission and exit dates of residents, and will also function reliably without the need for regular SARS-CoV-2 testing records.

Most care homes are unfamiliar with research practices and procedures, and whilst the use of routinely collected data can greatly reduce the need for prospective data collection, it is nonetheless preferable to combine these approaches in a clinical trial where possible. A greater amount of researcher-led primary data collection within the homes would have given us a better understanding regarding, for example, true testing and reporting levels. However, this would have required a substantial increase in resource need for the trial and would have made recruitment and retention of homes more difficult. Overall, our findings underscore the need for innovative approaches to care home research that enable the rapid generation of evidence to inform policy. This should include investment in training and capacity building for care home staff, so they have the knowledge, skills and ring-fenced time to support both intervention delivery and data collection. Such investment is required to bring the sector in line with other areas of health care delivery.

## Conclusions

Early cessation of the trial meant that we were unable to address our original research question, but progress was made in the development of a pragmatic approach for care home research to inform policy and practice. The trial was established in >80 care homes within 3 months, with rapid recruitment and intervention design facilitated by strong pre-existing

relationships between the research team, care providers and policymakers forged during the pandemic and by carers' familiarity with COVID-19 testing procedures. Linked data and testing infrastructure established during the COVID-19 pandemic made it possible to derive the majority of the trial outcomes from routinely collected data, reducing the need for primary data collection – a key barrier to trial participation.

The need for high-quality research in care homes to inform policy and practice is undisputed, as well as to respond to the needs of our ageing population and increasing levels of multi-morbidity. There are, however, major barriers to delivering clinical trials in this setting. Although the rapidly changing epidemiological and policy context precluded us from addressing our original research question, this trial represents an important step towards a more agile and participative model for 'real-time' care home research, which could be adapted to enable the efficient delivery of interventional studies in this setting beyond the COVID-19 pandemic.

## Supporting information

**S1 Data. Weekly aggregate data on primary and secondary outcomes at the level of each care home.**
(CSV)

**S2 Data. Specification of the variables included in S1 Data.**
(TXT)

**S1 Table. Simplifications required to achieve convergence of analysis models.**
(DOCX)

**S2 Table. Intervention effect estimates from mixed effects models without adjustment variables.**
(DOCX)

**S1 Appendix. Summary of analyses dropped from the Statistical Analysis Plan, and comparison of primary outcome event counts by data source.**
(DOCX)

**S1 File. Study Protocol.**
(PDF)

**S2 File. Randomisation plan.**
(PDF)

**S3 File. Statistical analysis plan.**
(PDF)

**S4 File. CONSORT checklist.**
(DOCX)

## Acknowledgments

We thank the staff and residents in the long-term care facilities who participated in this study and Mark Marshall at National Health Service (NHS) England who pseudonymised the electronic health records. The views expressed in this publication are those of the authors and not necessarily those of the NHS or the UK Health Security Agency.

We would like to acknowledge the support provided to the study by Trial Steering Committee (TSC) and Data Monitoring Committee (DMC) members Prof Alastair Hay (Chair TSC), Dr Jennifer Thompson (TSC), Dr Michael Larkin (TSC), Zoe Fry (TSC), Samantha Crawley (TSC), Prof Claire Hulme (TSC), Margaret Ogden (TSC), Prof Karla Hemming (Chair DMC), Dr Tania Kalsi (DMC) and Dr Terry Quinn (DMC).

## Author contributions

**Conceptualization:** Oliver Stirrup, James Blackstone, Natalie Adams, Maria Krutikov, Nick Freemantle, Adam Gordon, Martyn Regan, Martin Knapp, Lara Goscé, Catherine Henderson, Susan Hopkins, Arpana Verma, Jackie Cassell, Dorina Cadar, Tom Fowler, Andrew Copas, Paul Flowers, Laura Shallcross.

**Data curation:** Oliver Stirrup, Iona Cullen-Stephenson, Robert Fenner.

**Formal analysis:** Oliver Stirrup.

**Funding acquisition:** Oliver Stirrup, James Blackstone, Nick Freemantle, Adam Gordon, Martyn Regan, Martin Knapp, Lara Goscé, Catherine Henderson, Susan Hopkins, Arpana Verma, Jackie Cassell, Dorina Cadar, Tom Fowler, Andrew Copas, Paul Flowers, Laura Shallcross.

**Investigation:** Oliver Stirrup.

**Methodology:** Oliver Stirrup, Natalie Adams, Ruth Leiser, Adam Gordon, Jackie Cassell, Andrew Copas, Paul Flowers, Laura Shallcross.

**Project administration:** James Blackstone, Iona Cullen-Stephenson, Robert Fenner, Borscha Azmi, Tom Fowler.

**Resources:** Tom Fowler, Paul Flowers.

**Supervision:** Nick Freemantle, Andrew Copas, Paul Flowers, Laura Shallcross.

**Writing – original draft:** Oliver Stirrup, Laura Shallcross.

**Writing – review & editing:** Oliver Stirrup, James Blackstone, Iona Cullen-Stephenson, Robert Fenner, Natalie Adams, Ruth Leiser, Maria Krutikov, Borscha Azmi, Nick Freemantle, Adam Gordon, Martyn Regan, Martin Knapp, Lara Goscé, Catherine Henderson, Susan Hopkins, Arpana Verma, Jackie Cassell, Dorina Cadar, Tom Fowler, Andrew Copas, Paul Flowers, Laura Shallcross.

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
