## [Decision Letter · Decision Letter 0]

PONE-D-24-37992

VIVALDI-CT Shaping care home COVID-19 testing policy: A pragmatic cluster randomised controlled trial of asymptomatic testing compared to standard care in care home staff

PLOS ONE

Dear Dr. Stirrup,

Thank you for submitting your manuscript to PLOS ONE. After careful consideration, we have decided that your manuscript does not meet our criteria for publication and must therefore be rejected.

Specifically, based on the reviewers' comments, the concept behind this cluster randomised unblinded trial on COVID-19 testing in care homes is undoubtedly interesting and addresses an important population. However, due to the study being stopped early and underpowered, the results (or lack thereof) do not align with the priorities of the journal.

It may be more suitable to reframe the manuscript as a "lessons learnt" article or even a Viewpoint discussing how future studies could be designed and conducted. The authors seem to have valuable insights to contribute to the scientific community in this regard. Therefore, the manuscript, in its current form, is not deemed suitable for publication in the journal.

I am sorry that we cannot be more positive on this occasion, but hope that you appreciate the reasons for this decision.

Kind regards,

Sana Eybpoosh

Academic Editor

PLOS ONE

Reviewers' comments:

Reviewer's Responses to Questions

**Comments to the Author**

1. Is the manuscript technically sound, and do the data support the conclusions?

Reviewer #1: Partly

Reviewer #2: Yes

2. Has the statistical analysis been performed appropriately and rigorously?

Reviewer #1: Yes

Reviewer #2: Yes

3. Have the authors made all data underlying the findings in their manuscript fully available?

Reviewer #1: No

Reviewer #2: Yes

4. Is the manuscript presented in an intelligible fashion and written in standard English?

Reviewer #1: Yes

Reviewer #2: Yes

Reviewer #1: The concept behind this cluster randomised unblinded superiority trial of COVID-19 testing in care homes is certainly an interesting one, and is an important population to study. However, the trial was stopped early due to it being underpowered, and therefore the results (or lack thereof) of the study do not seem pertinent for PLOS.

It may be more appropriate to write the manuscript as a 'lessons learnt' or even a Viewpoint on how future studies should be conducted, as I think that the authors have a lot of lessons to contribute to the scientific community.

Reviewer #2: PLOS ONE PEER REVIEW

TITLE: VIVALDI-CT Shaping care home COVID-19 testing policy: A pragmatic cluster

randomised controlled trial of asymptomatic testing compared to standard care in care

home staff.

PONE-D-24-37992

SUMMARY AND OVERALL IMPRESSION

This is a methodologically novel study by Stirrup et al., primarily aimed at exploring the difference between future hospital admissions among the elderly who were asymptomatic vs symptomatic or contacts (per standard testing guidelines) for COVID-19 through testing, in order to direct national policy. The use of a Randomized Control Superiority Trial in this context was a perfect design. The study procedures and processes were clearly described and explained

Older people have been found to have a higher incidence of asymptomatic COVID-19, which also carried a significant risk of transmission. This study was therefore in keeping with the findings and recommendations by Barboza et al (2021). Unfortunately, the low recruitment and primary outcome rendered the study statistically less powered at an early stage due to the rapidly changing epidemiological trend. The authors have also been able to clearly outline the major limitations of the study rather than cooking results and making significance out of nothing.

MAJOR ISSUES

There was no major issue to warrant correction to the best of my knowledge, expertise and understanding of this study.

MINOR ISSUES

For the first sentence of the first paragraph under results, line 270 on page 21, I think the sentence should begin with a word, not a figure (i.e. Eighty-five (85) care homes…..)

MISCELLANEOUS COMMENTS

I hereby declare that I do not have any competing interest as far as this study is concerned.

**Do you want your identity to be public for this peer review?** For information about this choice, including consent withdrawal, please see our Privacy Policy

Reviewer #1: **Yes: ** Dr Michael Blank

Reviewer #2: **Yes: ** Dr. Abraham Kwadzo Ahiakpa

- - - - -

---

## [Author Response · Author response to Decision Letter 1]

22 Jan 2025

Oliver Stirrup, PhD

Institute for Global Health

University College London

London, WC1E 6JB

22nd January 2025

Dear Dr Eybpoosh

Thank you for agreeing to consider a revised version of our manuscript: “PONE-D-24-37992: VIVALDI-CT Shaping care home COVID-19 testing policy: A pragmatic cluster randomised controlled trial of asymptomatic testing compared to standard care in care home staff”. Please find attached a point-by-point response to your comments and to those of the Reviewers. We have made substantial adjustments to the text in order to emphasise the fact that the trial was discontinued before achieving its target sample size and to further explore the lessons that can be drawn from our experiences.

Yours sincerely,

Oliver Stirrup

On behalf of

Oliver Stirrup, James Blackstone, Iona Cullen-Stephenson, Robert Fenner, Natalie Adams, Ruth Leiser, Maria Krutikov, Borscha Azmi, Prof Nick Freemantle, Prof Adam Gordon, Prof Martyn Regan, Prof Martin Knapp, Lara Goscé, Catherine Henderson, Prof Susan Hopkins, Prof Arpana Verma, Prof Jackie Cassell, Dorina Cadar, Tom Fowler, Prof Andrew Copas, Prof Paul Flowers, and Prof Laura Shallcross

Comments of the Editor:

Specifically, based on the reviewers' comments, the concept behind this cluster randomised unblinded trial on COVID-19 testing in care homes is undoubtedly interesting and addresses an important population. However, due to the study being stopped early and underpowered, the results (or lack thereof) do not align with the priorities of the journal.

It may be more suitable to reframe the manuscript as a "lessons learnt" article or even a Viewpoint discussing how future studies could be designed and conducted. The authors seem to have valuable insights to contribute to the scientific community in this regard. Therefore, the manuscript, in its current form, is not deemed suitable for publication in the journal.

REPLY:

Authors’ response:

We feel that it is the responsibility of researchers to report trial results in full (as planned in the study Protocol) even in the event of premature discontinuation of the trial. This viewpoint can be well supported with reference to the established literature on good practice in the conduct of clinical trials. For example, in a review of the final outcomes of a set of registered clinical trials, Kasenda et al state “The nonpublication of results from discontinued—or from completed—RCTs represents a waste of valid data that could contribute to systematic reviews and meta-analyses”. Similarly, Speich et al wrote: “For example, if an RCT is discontinued due to slow participant recruitment before the planned sample size is reached, the trial is typically not sufficiently powered to answer the primary research question. The data, however, can still be useful in meta-analyses. Hence, it is crucial that all RCT results, including discontinued trials, are made available so that evidence is not lost and resource waste is minimized.”

We are very happy to expand further on ‘lessons learnt’ points in the Discussion of the manuscript, and to adjust the framing of our results. However, we do not feel that it is possible to do this meaningfully without full reporting of the data obtained. This is particularly pertinent for this study, given that we are demonstrating the feasibility of novel trial methodology using routine data collection within the setting of residential care homes and working towards the capacity to deliver responsive pragmatic trials in this setting within the timeframes required by policymakers. This is an area where more high-quality randomised research is urgently needed to inform policy, but where such studies are very difficult to set up due to challenges relating to recruitment of residents, staff workloads and lack of established research culture.

Once a trial has been registered and human participants recruited, it is the duty of the research team to report any and all results in full. Article 36 of the Declaration of Helsinki states that: “Researchers, authors, sponsors, editors, and publishers all have ethical obligations with regard to the publication and dissemination of the results of research. Researchers have a duty to make publicly available the results of their research on human participants and are accountable for the timeliness, completeness, and accuracy of their reports. All parties should adhere to accepted guidelines for ethical reporting. Negative and inconclusive as well as positive results must be published or otherwise made publicly available.” (https://www.wma.net/policies-post/wma-declaration-of-helsinki/). PLOS One explicitly specifies that all research involving human subjects should comply with the Declaration of Helsinki (https://journals.plos.org/plosone/s/human-subjects-research). We also note that the stated policy of PLOS ONE is “In keeping with our mission to publish all valid research, we consider negative and null results” (https://journals.plos.org/plosone/s/criteria-for-publication), and it is the stated philosophy of PLOS ONE that “we believe all rigorous science deserves to be published and should be discoverable” (https://journals.plos.org/plosone/s/journal-information ).

In response to the concerns of the Editor and Reviewer 1, we have added the following to the last paragraph of the Introduction section in order to adjust the expectations of anyone reading the manuscript:

“The trial ultimately did not recruit its target sample size of care homes due to shifts in the epidemiological and policy context in the UK. In this article we report on the trial outcomes obtained relating to our original aims, and also explore how this study could help inform the development of future cluster trials in residential care homes.”

We have also added the following paragraph to the Results to help frame their interpretation:

“Despite early cessation of the trial, we have reported full trial results as per the study’s Protocol and Statistical Analysis Plan. The trial was prospectively registered (ClinicalTrials.gov registration: NCT05639205) and we feel that full reporting of the study’s results is appropriate for transparent completion of the research process and to respect the contribution of research subjects (in line with the Declaration of Helsinki Article 36(25)). Furthermore, full reporting of trial outcomes demonstrates the feasibility of our novel use of routine health data for a randomised trial in the context of residential care homes, which allowed us to minimise the additional workload required from care staff of participating homes; we believe that development of this approach will be crucial to the delivery of greater numbers of randomised controlled trials in this setting, as well as enabling the responsiveness and compressed timelines required for research related to epidemic and pandemic diseases.”

We have now highlighted the better ascertainment of outcomes using routine data within the Results section “…this implies that the use of routine healthcare data has the potential to provide more reliable ascertainment of participant outcomes than trial-specific data collection in the setting of cluster trials in residential care homes.”

We have restructured the Discussion section, with an addition to the opening paragraph:

“Although the trial was not ultimately powered to meet its primary objective, we nonetheless feel that the results represent an important contribution to the development of large-scale and high-quality research in the setting of care homes for older people. We also hope that our results might contribute to future systematic reviews and meta-analyses, for which all evidence stemming from randomised studies can be useful(26, 27).”

We have also added an explicit and expanded ‘lessons learnt’ subsection, with the following new text:

“Lessons learnt

Our successful ascertainment of trial outcomes demonstrates the feasibility of our novel use of routine health data for a cluster randomised trial in the context of residential and nursing care homes. This is an area where more high-quality randomised research is urgently needed to inform policy, but where such studies are very difficult to set up due to difficulties relating to recruitment of care homes(37) and individual residents, and a lack of embedded research staff and infrastructure. A key barrier to the delivery of any kind of research in care homes is a lack of staff capacity to support study delivery because staff are overstretched with high vacancy rates, and research is not typically part of the role for front-line carers. In England, studies conducted in the NHS benefit from support and direct financial incentives for recruitment via the NIHR Research Delivery Network, but this infrastructure is undeveloped in care homes. Cluster trials typically require large number of individual participants(38), making it very challenging to collect all of the data needed for a trial. The creation of a reliable pipeline for routine healthcare data of care home residents is therefore crucial for the development of future large-scale trials in this setting in the UK.

The preceding Vivaldi observational study (ISRCTN 14447421) began as a SARS-CoV-2 sero-prevalence study(18), but we realised that there was a unique opportunity to support the COVID-19 response in care homes by integrating routinely collected individual-level testing and vaccination data from residents and staff, linked to healthcare outcomes. This made it possible — for the first time ever in England — to establish an accurate database of care home residents and staff that could be used for COVID-19 research to directly inform policy in near real-time. The resulting cohort study included >300 care homes and individual-level data from >70,000 staff and residents, which we used to generate evidence on disease burden and severity(36), variants(39) and vaccine effectiveness(12, 21). For the present trial, we re-purposed the care home network and data infrastructure established through Vivaldi to rapidly set-up a pragmatic, cluster randomised trial.

We are developing a new pilot study, Vivaldi Social Care(40), with the goal to make it easier for care homes to participate in research studies (by keeping the workload associated with study participation to a minimum), to increase research impact and efficiency by enabling studies to progress at pace and scale, and to deliver research in partnership with the care sector. We are expanding the Vivaldi network to c.1500 care homes, and establishing new processes to access infection-related routinely collected data from residents in these homes. Through collaboration with care sector organisations (e.g. The Outstanding Society, Care England and others), and extensive engagement with residents, relatives, charities and care homes staff, we are gradually changing the culture and understanding of research in homes that are participating in this project. A key technical development for the Vivaldi Social Care project(40) is that the data pipeline will be based on identification codes (i.e. NHS numbers) of residents exported daily directly from the electronic record systems of participating care homes; this will provide an improved level of accuracy regarding admission and exit dates of residents, and will also function reliably without the need for regular SARS-CoV-2 testing records.”

REFERENCES

Kasenda B, von Elm E, You J, Blümle A et al. Prevalence, characteristics, and publication of discontinued randomized trials. JAMA 2014 12;311(10):1045-51. doi: 10.1001/jama.2014.1361.

Speich B, Gryaznov D, Busse JW et al. Nonregistration, discontinuation, and nonpublication of randomized trials: A repeated metaresearch analysis. PLoS Med 2022;19(4):e1003980. doi: 10.1371/journal.pmed.1003980.

Reviewer Comments:

Reviewer #1: The concept behind this cluster randomised unblinded superiority trial of COVID-19 testing in care homes is certainly an interesting one, and is an important population to study. However, the trial was stopped early due to it being underpowered, and therefore the results (or lack thereof) of the study do not seem pertinent for PLOS.

It may be more appropriate to write the manuscript as a 'lessons learnt' or even a Viewpoint on how future studies should be conducted, as I think that the authors have a lot of lessons to contribute to the scientific community.

REPLY:

As detailed in our reply to the Editor, we feel that it is appropriate to fully report trial outcomes even in the event of below-target recruitment of care homes to the study. We have adapted our framing of the trial results in the Introduction and Results and have substantially expanded on the ‘lessons learnt’ aspects of the Discussion, as detailed above, whilst retaining reporting of trial data.

Reviewer #2: PLOS ONE PEER REVIEW

TITLE: VIVALDI-CT Shaping care home COVID-19 testing policy: A pragmatic cluster

randomised controlled trial of asymptomatic testing compared to standard care in care

home staff.

PONE-D-24-37992

SUMMARY AND OVERALL IMPRESSION

This is a methodologically novel study by Stirrup et al., primarily aimed at exploring the difference between future hospital admissions among the elderly who were asymptomatic vs symptomatic or contacts (per standard testing guidelines) for COVID-19 through testing, in order to direct national policy. The use of a Randomized Control Superiority Trial in this context was a perfect design. The study procedures and processes were clearly described and explained

REPLY:

Thank you for the positive comments regarding the study design.

Older people have been found to have a higher incidence of asymptomatic COVID-19, which also carried a significant risk of transmission. This study was therefore in keeping with the findings and recommendations by Barboza et al (2021). Unfortunately, the low recruitment and primary outcome rendered the study statistically less powered at an early stage due to the rapidly changing epidemiological trend. The authors have also been able to clearly outline the major limitations of the study rather than cooking results and making significance out of nothing.

REPLY:

We feel that we are in agreement with this Reviewer that it is best to clearly describe the limitations of the study in the context of full and transparent reporting, rather than attempting to spin a more positive narrative based on secondary outcomes.

MAJOR ISSUES

There was no major issue to warrant correction to the best of my knowledge, expertise and understanding of this study.

MINOR ISSUES

For the first sentence of the first paragraph under results, line 270 on page 21, I think the sentence should begin with a word, not a figure (i.e. Eighty-five (85) care homes…..)

REPLY:

Thank you for flagging this, we would be very happy to make this amendment.

MISCELLANEOUS COMMENTS

I hereby declare that I do not have any competing interest as far as this study is concerned

---

## [Decision Letter · Decision Letter 1]

VIVALDI-CT Shaping care home COVID-19 testing policy: A pragmatic cluster randomised controlled trial of asymptomatic testing compared to standard care in care home staff

PONE-D-24-37992R1

Dear Dr. Stirrup,

We’re pleased to inform you that your manuscript has been judged scientifically suitable for publication and will be formally accepted for publication once it meets all outstanding technical requirements.

Kind regards,

Sascha Köpke

Academic Editor

PLOS ONE

Additional Editor Comments (optional):

Reviewers' comments:

Reviewer's Responses to Questions

**Comments to the Author**

Reviewer #2: All comments have been addressed

Reviewer #3: All comments have been addressed

Reviewer #4: All comments have been addressed

2. Is the manuscript technically sound, and do the data support the conclusions?

Reviewer #2: Yes

Reviewer #3: Partly

Reviewer #4: Yes

3. Has the statistical analysis been performed appropriately and rigorously?

Reviewer #2: Yes

Reviewer #3: Yes

Reviewer #4: Yes

4. Have the authors made all data underlying the findings in their manuscript fully available?

Reviewer #2: Yes

Reviewer #3: Yes

Reviewer #4: Yes

5. Is the manuscript presented in an intelligible fashion and written in standard English?

Reviewer #2: Yes

Reviewer #3: Yes

Reviewer #4: Yes

Reviewer #2: PLOS ONE PEER REVIEW

TITLE: VIVALDI-CT Shaping care home COVID-19 testing policy: A pragmatic cluster

randomized controlled trial of asymptomatic testing compared to standard care in care

home staff.

PONE-D-24-37992

-

Response

Upon review of the revised manuscript, the authors have duly addressed my earlier concerns, and I currently do not have any major or minor issues with this work. I therefore, on my side, leave the rest to the discretion of the Editor.

Thank you.

Reviewer #3: Previous questions have been addressed. Nil further to add

Reviewer #4: I appreciate that authors addressed the comments from my previous review. I have read the revised manuscript and concluded that the paper is suitable for publication in this journal.

**Do you want your identity to be public for this peer review?** For information about this choice, including consent withdrawal, please see our Privacy Policy

Reviewer #2: **Yes: ** Dr Abraham Kwadzo Ahiakpa

Reviewer #3: No

Reviewer #4: No

---

## [Editor Report · Acceptance letter]

PONE-D-24-37992R1

PLOS ONE

Dear Dr. Stirrup,

I'm pleased to inform you that your manuscript has been deemed suitable for publication in PLOS ONE. Congratulations! Your manuscript is now being handed over to our production team.

Kind regards,

on behalf of

Professor Sascha Köpke

Academic Editor

PLOS ONE